# Exploring Hikikomori-like idiom of distress a year into the SARS-CoV-2 pandemic in Oman: Factorial validity of the 25-item Hikikomori Questionnaire, prevalence and associated factors

**Nasser Al-Sibani[1]\*, Moon Fai Chan[2], Salim Al-Huseini[3], Nutaila Al Kharusi[1], Gilles J. Guillemin[4], Mohammed Al-Abri[5], Aishwarya Ganesh[1], Yousuf Al Hasani[6], Samir Al-Adawi[1]\***

1 Department of Behavioural Medicine, College of Medicine & Health Sciences, Sultan Qaboos University, Al-Khoudh, Muscat, Oman, 2 Department of Family Medicine and Public Health, College of Medicine & Health Sciences, Sultan Qaboos University, Al-Khoudh, Muscat, Oman, 3 Department of Psychiatry, Al Masarrah Hospital, Ministry of Health, Muscat, Oman, 4 Neuroinflammation Group, Macquarie Medical School, Macquarie University, Sydney, New South Wales, Australia, 5 Clinical Physiology, Sultan Qaboos University, Al-Khoudh, Muscat, Oman, 6 Organizational Development & Education, Curative Services, Directorate General of Medical Services, Muscat, Oman

\* nsibani@squ.edu.om (NS); samir.al-adawi@fulbrightmail.org (SA)

**Data Availability Statement:** All relevant data are within the paper.

## Abstract

### Background

Existing literature that examines the Hikikomori-like idiom of distress (HLID) initially labelled this phenomenon as a culture-bound syndrome. However, the increasing number of reports from other parts of the world suggest that HLID could instead be a culture-reactive idiom of distress, originating from biopsychosocial disequilibrium and cultural upheaval. The impact of imposed social withdrawal due to the Coronavirus 2 severe acute respiratory syndrome (SARS-CoV-2) pandemic on the growing prevalence of HLID has not yet been adequately explored. Aims: The first objective is to examine the factorial validity of the 25-item *Hikikomori Questionnaire* (HQ-25) in Oman. Second, we aimed to investigate the prevalence of HLID following the lifting of SARS-COV-2 restrictions in the Arabian Gulf country of Oman. The third related objective is to examine factors associated with the variation of HLID.

### Methods

This cross-sectional online survey was widely distributed across the diverse socio-demographic population residing in Oman. After the validation of the questionnaire and the factorial validity, the Arabic version of HQ-25 was used to explore the prevalence and factors associated with HLID.

**Funding:** The author(s) received no specific funding for this work.

**Competing interests:** The authors have declared that no competing interests exist.

## Results

A total of 1529 participants were included in the study (response rate = 76.5%), of whom 44% (n = 673) expressed HLID. We randomly split it into half, one for exploratory factor analysis (EFA) (n = 764) and the other half for confirmatory factor analysis (CFA) (n = 765). From the EFA results, a three-factor model was found for the Omani version of HQ-25, which represented 52.87% of the variance with good internal reliability (Overall Cronbach's: 0.92; Socialisation: 0.92; Isolation: 0.84; Emotional support: 0.73). The CFA results report acceptable goodness-of-fit indices ($\chi^2$/df = 17.92, p < .001; CFI = 0.90; TLI = 0.95; RMSEA = 0.04, 95% CI 0.02–0.07; SRMR = 0.05) of the three-factor model of the collected samples. All 1529 data were used in the respondents. The results of the logistic regression showed that married marital status (OR = 1.51, 95% CI: 1.12–2.03, p = 0.007), older age (OR = 0.97, 95% CI: 0.95–0.99, p = 0.008), living in an urban area (OR = 0.71, 95% CI: 0.56–0.91, p = 0.006), unemployed occupational status (OR = 1.72, 95% CI: 1.30–2.88, p < .001), screen time (7+ hours vs. <4 hours: OR = 4.50, 95% CI: 3.16–6.41, p < .001; 4–7 hours vs. <4 hours: OR = 2.10, 95% CI: 1.61–2.70, p < .001), history of mental illness (OR = 3.70, 95% CI: 2.29–5.91, p < .001), and adverse childhood experiences (OR = 2.60, 95% CI: 2.03–3.20, p < .001) were significant risk factors for HLID.

## Conclusion

The factorial validity of the HQ-25 performed in this study appears to mirror the previously reported 3-factor structures. Some of the associated factors appear to support and, conversely, dissent from the findings of previous studies. These are discussed in terms of the attributions of the SARS-COV-2 pandemic, demographic trends in Oman, and sociocultural factors specific to the region of interest.

## 1 Introduction

The existing literature on people struggling with social isolation and confinement often discusses diagnoses such as pathological social withdrawal [1], 'social diseases' [2], and avoidant personality disorder [3]. With the increasing emergence of reports of prolonged home confinement and complete social isolation in Japan, terms such as Hikikomori syndrome and Hikikomori disorder have come to the fore [4].

Since the first reports of *hikikomori* emerged from Japan [5–7], this phenomenon of severe social withdrawal among Japanese youth has been widely discussed as a culture-bound syndrome [8, 9]. Symptoms have been attributed to a wide range of sociocultural factors specific to Japan, such as culturally favourable views on traditional hermitism and asceticism [10], the psychoanalytic concept of object relations often applied to Japanese child-rearing practises [11], as well as cultural patterning and rigid societal expectations that favour conformism that is intimately tied to the collective mind set of Japan. Japanese demographic trends, social emphasis on education, as well as economic and labour factors, have all been suggested to support the theory that *hikikomori* is a strictly culture-bound syndrome [12–14]. However, with *hikikomori* being increasingly reported from different parts of the world, the pendulum of discussion has swung towards the possibility that it is a culture-reactive phenomenon instead [14–18]. It has also been theorised that *hikikomori* is less a clinical syndrome than an idiom of distress, so the term Hikikomori-like idiom of distress (HLID) has been used for the present study [4, 19]. The idiom of

distress is a term used to describe cultural expressions or behaviours that people use to communicate their psychological distress or suffering. In the context of socially sanctioned withdrawal due to the Coronavirus 2 of Severe Acute Respiratory Syndrome (SARS-COV-2), the idiom of distress associated with hikikomori may manifest through avoidance of school or work, seclusion in one's room, and social withdrawal. Thus, the term "idiom of distress" helps to recognize that social withdrawal intimately linked to something akin to hikikomori is a culturally specific response to perceived distress. The SARS-COV-2 pandemic has caused widespread social distancing and isolation measures, which may have some similarities to the characteristics of hikikomori. The proposed culture is reactive to social withdrawal or isolation and the socially sanctioned idiom of distress has received scant attention in the existing literature. The SARS-COV-2 pandemic may exacerbate existing temperaments such as those characterised by shy-inhibited individuals. Consistent with this view, Gray & McNaughton [20] have suggested two distinct systems in the brain that regulate behaviour and motivation. The Behavioral Inhibition System(BIS) is responsible for detecting and responding to potential threats or punishment. It is associated with feelings of anxiety and avoidance behaviour. On the other hand, the Behavioral Activation System (BAS) is responsible for detecting and responding to potential rewards or positive outcomes. Both systems work together to influence an individual's personality, emotional reactivity, and behaviour in different situations. Therefore, it is hypothesised that phenotypes such as BIS could lead individuals to adopt HLID.

This culture-reactive hypothesis is aligned with the diathesis-stress model of psychopathology, which asserts that a person's genetic or biological predisposition influences the probability of developing distress when a particular stressor is present [21]. The diathesis-stress model also appears to be supported by the recent tribulations of the SARS-COV-2 pandemic [22]. With the emergence of the pandemic, when lockdowns and social isolation were promoted to curb infection spread, there has been heightened reporting of the proliferation of loneliness and the development of something akin to HLID [23–25]. There is increased speculation that this externally enforced social distancing may trigger vulnerable individuals, such as those with 'shy-inhibited temperaments' as postulated by the diathesis-stress model, to develop self-imposed social distancing and withdrawal behaviours even without external influence following the pandemic [26]. To date, although there have been many theoretical studies on HLID, there are few epidemiological studies conducted outside of Japan [12, 18, 27, 28], and none have reflected the pandemic situation in emerging economies.

The culture-reactive hypothesis for HLID appears to echo studies by Kato and Kanba [6], Norasakkunkit and Uchida [2], and Ryder et al. [3], suggesting that the influx and proliferation of the modern ethos of self-determination and individualistic social patterning in culturally interdependent or communal societies tend to challenge their integrity. It has been postulated that when such divergent cultural ideologies intertwine, this is likely to act as a strong harbinger of reactionary maladaptive coping strategies such as HLID. The increasing prevalence of HLID has recently culminated in a 'silent epidemic' in several East Asian countries that embrace collectivistic social patterns and Confucianism (or the importance of individual morality) [1, 29]. It remains to be studied whether such idioms of distress can be found in other similar societies where collective mindsets prevail amid acculturation. Within the culture-reactive hypothesis, in addition to the collectivistic versus individualistic dichotomy, Orsolini et al. [30] have also hypothesized that technological revolutions that are rapidly changing modes of living have, in turn, eroded 'family structure and intra-familial dynamics' (p. 2) in societies that were thought to already be out of sync with collective patternings, such as Italy, where HLID is increasingly reported [31]. According to this point of view, post-modern societies can trigger specific idioms of distress that previously appeared to be 'culture bound' [2, 32].

To further explore this culture-reaction contention, more studies are required to explore the prevalence of HLID in emerging economies, such as countries of the Arabian Gulf. Oman, a country with an estimated 4 million population, is located at the southern tip of the Arabian Peninsula [33]. Society in Arab Gulf countries such as Oman also exhibits collectivist social patterning, as well as socioeconomic changes fostered by the industrial exploitation of hydro-carbons, tourism, and trans-shipment. According to Smith [34], ". . .money from oil. . . has brought the Omanis through development that took a thousand years in Europe in less than 20 years" (p. 297). The *Human Development Index* (HDI) has lauded Oman as the 'most improved nation' in the world, in terms of the previous decade [35]. The per capita income in Oman has grown rapidly, which in turn has increased material prosperity. Demographically, Oman appears to be a society in transition and currently at 'stage three'—implying, using the prism of demographic transition, that the country has taken on several core features of a post-modern society [36]. The question remains whether these features of collectivism can coexist with the inherent societal scepticism that is characteristic of postmodern societies, without lay-ing the groundwork for culture-reactive idioms of distress like HLID. Some preliminary epide-miological studies have suggested that maladjustment and overt psychiatric disorders, including social withdrawal, are becoming increasingly common in Oman [9, 37–39]. Using subclinical indices developed for tapping into shy-inhibited and individuals marked with avoi-dant personalities, known in the local vernacular as *Khajuul*—a term that encapsulates social anxiety disorder, or *Taijin Kyofusho* in international psychiatric nosology, Ambusaidi et al. [40] reported that the prevalence of this subclinical disposition affected 45.9% of the respon-dents surveyed in the Omani community. This figure echoes previous studies conducted among secondary school and tertiary education students [41, 42]. It remains to be established how such cases of HLID-adjacent disorders fare following the lifting of restrictions placed due to the SARS-COV-2 pandemic, and to what extent these reports echo those found in the extant literature. The SARS-COV-2 outbreak was officially declared a pandemic in March 2020 [43]. During the different waves of the pandemic, the Omani government adopted various WHO guidelines to prevent the spread of SARS-COV-2 among the population. During the peak of SARS-COV-2 infection rates, the Oman border was closed and, if applicable, highly affected regions were cordoned off. This was enforced through joint checkpoints to curb those who may flout the social distancing rules. Employers mandated that only 30% of the workforce be physically present at their place of work. On the other hand, all classes in schools, universities, and other educational institutions were closed. In general, all public places (cinemas, gyms, sports clubs, barber shops, beaches), including shops and shopping malls, apart from food and medical outlets were closed. Mosques and other religious places were also closed and the two official holidays celebrated within Islam (Eid al-Fitr and Eid al-Adha) were marked with more stringent rules of social isolation and curfew. In March 2022, at the time of this study, most government-imposed social distancing regulations were lifted in response to a relatively mini-mal number of infection reports in Oman. Exploring the prevalence of self-imposed social withdrawal practises so soon after the loosening of these restrictions could result in false infla-tion in the number of people who remain 'housebound' due to their fears or a sense of civic responsibility with respect to the spread of infection. However, this alone can be considered a form of self-imposed isolation, which in turn increases loneliness and the risk of adverse men-tal health outcomes and HLID [28]. Therefore, it is important to assess the prevalence of this persistent behaviour for further contextualisation. Some preliminary studies have begun to shed light on some of the possible biomarkers of HLID [44, 45]. There are various HLID assessment tools available to the public including the *Hikikomori Behavior Checklist* [46], *Adaptive Behaviors Scale for Hikikomori* [47], *Hikikomori Assessment Interview* [48], *Not in Employment Education or Training (NEET)/Hikikomori Risk Scale* [2], *Hikikomori Risk*

*Inventory* [49], but the *Hikikomori Questionnaire* HQ-25 was purposefully chosen for the present study for two interrelated reasons. First, *hikikomori* does not just imply physical distancing of an individual from others but also involves their abilities and preferences for developing long-lasting, meaningful relationships through trust and communication. All these aspects of social relationships, socialisation, isolation, and emotional support, are scrutinised viaHQ-25 [50]. Second, among many varied tools that connect to HLID, the HQ-25 has been validated in various populations around the world. To date, none of these HLID assessment tools has been used in the Arabian Gulf population. Such an undertaking will further help to unravel whether HLID is a culture-bound or culture-reactive phenomenon.

This study aims to use exploratory factor analysis (EFA) and confirmatory factor analysis (CFA) methods to confirm the factorial validity of the *Hikikomori Questionnaire* (HQ-25), one of the most commonly used measures to assess the socialisation, isolation, and emotional support aspects of HLID [50, 51]. Establishing the taxonomy and phenotypes of HLID has the potential to lead to early recognition and prompt management to prevent its entry into chronic and refractory distress. The second and third objectives were to explore the prevalence of HLID in Oman and associated predictors. Related objectives are to advance the discourse on culture-reactive versus culture-specific idioms of distress from the Arabian Gulf population, as well as debate on universalist versus relativist mental health nosology to appreciate the effect of globalisation amid pluralism in Oman.

## 2 Materials and methods

### 2.1. Study design, participants, and data collection

Data collection was carried out over 2 weeks (in March 2022) among the general population of Oman. All participants 18 years or older were asked to complete an online survey. An electronically signed, printable consent form was sent to the invitees specifying that they needed to give their consent before opening the study survey. All participants were informed that their participation would remain confidential and voluntary and that they could withdraw from the study at any time. The survey was anonymous (no collection of personal data), and confidentiality of information was ensured.

### 2.2. Power analysis

The power analysis of this study was based on one of the main outcomes, the prevalence rates of HLID in Oman. If this study expected the prevalence rate measured by the HQ-25 to range from 8.1% [32] to 9.5% [46, 47, 50], with a ±1.5% precision at a 95% confidence interval, the required respondent's number ranged from 1271 to 1468. Taking into account a 25% nonresponding rate, this study required at least 1695 to 1958 respondents. Therefore, the online survey has targeted 2000 potential respondents to meet this requirement. The required sample size was calculated using the EpiTools online program [52].

### 2.3. Inclusion and exclusion criteria

The inclusion criteria included all adult Omanis (age > 18 years) who can read and write Arabic / English, who could provide consent using the provided consent form, those who have reliable internet access and who self-imposed social withdrawal after the onset of the SARS--CoV-2 pandemic. Exclusion included minors (age <18 years), people who could not read and write Arabic / English or did not give their electronic consent, those who do not have reliable Internet access, and those with a diagnosis of a psychiatric disorder that better accounts for social withdrawal (e.g., schizophrenia, autism spectrum disorder).

## 2.4. Sampling method

Data collection was carried out through an online survey using a convenience sampling method using the snowball technique. To reach all strata of society, the URL of the study survey was distributed through popular social media platforms such as Twitter, WhatsApp, Instagram and Facebook. Oman has increasingly witnessed the growth of social influencers, some of them known to act as mental health influencers or 'mental health gurus'. Being a snowball technique, mental health influencers were contacted to help cascade the present survey. A study has suggested the heuristic value of using social influencers to improve participation rates in online mental health surveys [52]. On the cover page, each recipient who received the link was encouraged to send the link to other people in his or her network. Until the sample size was reached, the initial contacts were recalled at least twice within the prescribed study time—two weeks. To prevent a participant from responding more than once, the survey asked participants to provide their unique email addresses when completing the questionnaire, which allowed the authors to verify that each response came from a unique participant. Additionally, the authors used cookies to prevent participants from accessing the questionnaire multiple times from the same device.

## 2.5. Outcome measures

**2.5.1 Sociodemographic and risk and clinical variables.** The study survey contained two parts to be detailed in the following in tandem. The first part, as shown in Table 1, constituted sociodemographic information that included sex, marital status, age, and level of education. For theoretical reasons, the urban vs. rural dichotomy was also explored. In countries with emerging economies such as Oman, previous studies have suggested that the prevalence of maladaptation tends to strongly depend on whether one resides in a rural or urban setting. The split between what constitutes rural and urban in the context of Oman has been established in other studies, with the same categorisations used presently [40]. Other variables explored include socioeconomic status, whether the participant was currently living alone or otherwise, an occupation that included student, unemployed, and employed, and average daily screen time or time spent on the Internet / social networks. Screen time assessment is made on the rationale that HLID has been strongly associated with 'internet addiction' [53]. To analyse the associated clinical variables and risk factors, the present study includes items to determine whether the participants had a prior diagnosis of mental illness ("have had any current or ongoing diagnoses of mental illness?"), had a family history of mental illness, or a history of adverse childhood experiences (ACE) such as physical, sexual, or emotional abuse, smoking, or drinking alcohol. These were evaluated to help estimate the prevalence of primary HLID and secondary HLID with comorbid mental illness in the sample [54].

**2.5.2 Self-reported HLID.** To access HLID, the *Hikikomori Questionnaire (HQ-25)* was used [50]. A self-administered instrument for assessing HLID, HQ-25, was used to tap into behavioural patterns that a growing body of research has suggested constitute the hallmark of HLID, including lack of social connection, active social isolation or withdrawal behaviour, avoidance of social contact and a sense of alienation from society [18, 55, 56]. HQ-25 is a self-reported instrument of Likert types (0 = strongly disagree; 1 = somewhat disagree; 2 = neither agree nor disagree; 3 = somewhat agree; 4 = strongly agree). The HQ-25 score ranges from 0–100, with higher scores indicating a higher severity of hikikomori-like symptoms. Teo et al. [50] reported internal consistency, test-retest reliability, and convergence validity (95% confidence interval, 0.80–0.92) of the Japanese version of HQ-25. A cut-off score of 42 (out of 100) has been widely used in different populations [32, 50, 51]. For the 42 present study, the cutoff

**Table 1. Characteristics of the study respondents (n = 1529).**

| Characteristics | n (%) |
|---|---|
| **_Demographics_** | |
| Gender | |
| Female | 1035 (67.7) |
| Male | 494 (32.3) |
| Marital status | |
| Single | 553 (36.2) |
| Married | 939 (61.4) |
| Divorced/widow | 37 (2.4) |
| Age (years) | |
| Mean±SD | 31.2±7.7 |
| Median [Range] | 31.0 [19.0–45.0] |
| Education level | |
| High school/below | 221 (14.5) |
| University/above | 1308 (85.5) |
| Place of living | |
| Urban | 580 (37.9) |
| Rural | 949 (62.1) |
| Socio-economic Status | |
| Upper | 62 (4.1) |
| Middle | 560 (36.6) |
| Low | 907 (59.3) |
| Living arrangements/Living alone | |
| Yes | 254 (16.6) |
| No | 1275 (83.4) |
| Occupation | |
| Employed | 858 (56.1) |
| Unemployed | 671 (43.9) |
| Screen time (hours/day) | |
| >7 | 247 (16.2) |
| 4–7 | 585 (38.3) |
| <4 | 697 (45.6) |
| **_Clinical variables_** | |
| Having mental illness | |
| No | 1413 (92.4) |
| Yes | 116 (7.6) |
| Depression | 47 (40.5) |
| Anxiety | 34 (29.3) |
| Social phobia | 20 (17.2) |
| Obsessive-compulsive disorder | 6 (5.2) |
| Personality disorder | 2 (1.7) |
| Others (e.g., Psychosis) | 7 (6.0) |
| Family history of mental illness | |
| Yes | 303 (19.8) |
| No | 1226 (80.2) |
| Adverse childhood Experience (physical, sexual/emotional abuse) | |
| Yes | 913 (59.7) |
| No | 616 (40.3) |

(*Continued*)

**Table 1.** (Continued)

| Characteristics | n (%) |
|---|---|
| Smoking | |
| Yes | 76 (5.0) |
| No | 1453 (95.0) |
| Drinking alcohol | |
| Yes | 22 (1.4) |
| No | 1507 (98.6) |

established by Teo et al. [50] was used, which was reported to have a sensitivity of 94%, a specificity of 61%, and a positive predictive value of 17%.

To date, there is no Arabic version of HQ-25. In addition to translating the HQ-25 questionnaire originally reported by Teo et al. [50] into Arabic, using the publicly accessible English version with permission from Dr. Alan Teo's website (https://www.dralanteo.com/services-2), this study also investigated its factorial validity. The established protocols, back-translation, were used to translate HQ-25 into the Omani dialect. Academicians in cross-cultural studies or those who practised behavioural sciences/psychology/psychiatry and were well versed in both spoken and written English and Arabic were asked to participate in the translation of HQ-25. According to established protocol, a team was asked to translate the original version into Arabic. Once the Arabic version was established and a consensus was reached, another academic team was asked to translate the Arabic version of HQ-25 into English. The two teams met online to review the items and content of the questionnaire. If there was disagreement, further changes were made to ensure total agreement on each item between the original and the back-translated English.

The newly drafted HQ-25 in Omani dialect was piloted among students attending behavioural lectures, where they were asked to fill out the questionnaire. The best practise of confidentiality was also maintained for the consenting students (n = 45). Among the Omani version of HQ-25, 23 of them were deemed to have been meaningfully filled by volunteers. The internal validity of the newly translated HQ-25 was calculated to have a Cronbach alpha of 0.75.

## 2.6. Ethical approval

Ethical approval was granted by the College of Medicine and Health Sciences at Sultan Qaboos University, Muscat, Oman (SQU-EC370/2021, MREC# 2393). Written informed consent was obtained from all participants. The study was carried out according to the principles of the Declaration of Helsinki with respect to ethical human research, including confidentiality, privacy, and data management. All participants were informed that their participation would remain confidential and voluntary, that the collected data would be aggregated, and that they could withdraw their participation and their data from the study at any time.

## 2.7. Statistical analysis

Descriptive statistics (e.g., mean, median, frequency) were used to describe the profile of the study respondents. The Arabic version of HQ-25 was then tested using exploratory factor analysis (EFA) and confirmatory factor analysis (CFA) methods to confirm its factorial validity. The data, which consist of 1529 observations, were randomly divided into two halves. One half, comprising 764 observations, was used for exploratory factor analysis (EFA), while the other half, consisting of 765 observations, was used for confirmatory factor analysis (CFA).

**2.7.1 Factorial validity of the Arabic version of HQ-25.** Exploratory factor analysis (EFA) was performed to determine whether responses to the questions could be grouped into scales that measure a common factor. The EFA was performed using IBM SPSS 23 (IBM Corp, Armonk, NY, USA). The principal components were used as the initial extraction method for factor analysis. The factor identification was based on the eigenvalue (> 1) and the scree plot. For comparison purposes, an oblique rotation (promax) with factor loadings ≥ 0.40 was used, as suggested by Teo et al. [50]. Subsequently, the factors identified in the EFA were analyzed for internal consistency (reliability). The Cronbach's alpha of each factor was considered acceptable if the value was > 0.7 [57, 58]. Furthermore, the Bartlett sphericity test and the Kaiser-Meyer-Olkin (KMO) sampling adequacy were used to determine the suitability of the suggested model [59].

Confirmatory factor analysis (CFA) was performed to test the factorial structure of the Arabic version. Using maximum likelihood estimation, the CFA was calculated with JASP 0.17.1 for Windows [60]. For CFA, the goodness of fit was tested with a $\chi^2$ divided by the degree of freedom [61–63]. A normed < 2 indicates a good fit of the model, and < 3 is an acceptable fit. Based on recommendations from previous studies [62], this study chose five indices for CFA: Standardized root mean square residual (SRMR), Root Mean Square Error of approximation (RMSEA), Comparative Fit Index (CFI), Tucker-Lewis index (TLI), and Goodness of Fit Index (GFI). SRMR and RMSEA are both residuals-based, absolute fit measures. The combination of RMSEA and SRMR is valuable because RMSEA is sensitive to misspecification of factor loadings, whereas SRMR is sensitive to misspecification of factor covariances. The latent and measurement model is well specified if both indices are accepted. Furthermore, the RMSEA is associated with a confidence interval. RMSEA values < 0.06 have been found to indicate a good fit. SRMR values should be < 0.08. CFI and TLI are additional relative fit measures, and values > 0.95 indicate a good model-data fit [62, 64].

After validation, the Arabic version of the HQ-25 score was used to explore the factors associated with the presence of HLID. Univariate and multivariate logistics analyses were used to examine which factors were associated with the presence of HLID compared to non-HLID subjects. The model was assessed using the Hosmer and Lemeshow test, Cox and Snell $R^2$, sensitivity and specificity. Associations and collinearity were examined using the odds ratio (OR) and the variance inflation factor (VIF), respectively. For each factor, if VIF > 5, it will be considered problematic [65]. The analyses were performed using IBM SPSS Statistics for Windows (version 23.0., IBM Corp., Armonk, NY, USA). A p-value less than 0.05 (2-tailed tests) was considered statistically significant.

# 3 Results

## 3.1. Descriptive results

In Table 1, a total of 1529 respondents participated in the study (response rate = 76.5%), 67.7% of the participants identified as women and 32.3% as men. The mean age of all participants was 31.2 ± 7.7, with minimum and maximum ages equal to 19 and 45 years, respectively. Most of the participants were married (n = 939, 61.4%), received university education (n = 1308, 85.5%), and lived with their family members (n = 1275, 83.4%). Almost half were employed (n = 858, 56.1%) and presently defined as the low socio-economic class (n = 907, 59.3%). A small number of participants had smoking and drinking alcohol habits, and in terms of screen time per day, 16.2% spent 'more than 7 hours', 38.3% spent '4–7 hours', and 45.6% spent 'less than 4 hours'. Among clinical risk factors, 7.6% (n = 116) reported having an associated mental illness with 40.5% (n = 47), 29.3% (n = 34), and 17.2% (n-20) reporting depression, anxiety,

and social phobia, respectively. 19.8% (n = 303) have a family history of mental illness and 69.7% (n = 913) have experienced adverse childhood experiences (ACE).

## 3.2. Exploratory factor analysis

During data analysis, HQ-25 items were found to be normally distributed in skewness (range from -0.30 to 0.84) and kurtosis (range from -1.14 to -0.21), so the 25 items were used in the EPA and CPA for further analysis (Table 2). The EFA was conducted with 764 participants. A 3-factor structure with eigenvalues of 9.53, 2.40, and 1.29 in factors 1, 2, and 3, respectively, while the remaining 22 factors with eigenvalues were below 1.0. Together, the three-factor model (factor 1: 38.12%, factor 2: 9.61%, factor 3: 5.14%) represented 52.87% of the variance. The KMO test was used to ensure the adequacy of the participants in conducting EFA (KMO = 0.95). Bartlett's test examined the items' correlation ($\chi^2$ = 9087.57, $df$ = 300, p < .001). Given the results and the significance level, performing EFA on HQ-25 was acceptable. To examine the internal consistency (reliability) of HQ-25, the overall Cronbach alpha was

**Table 2. Distribution of each item and comparison of factor loadings and internal consistency for the *Hikikomori Questionnaire*-25 items (HQ-25).**

| Exploratory Factor Analysis | Mean (SD) | Skewness | Kurtosis | Arabic | Japanese^ |
|---|---|---|---|---|---|
| *Factor 1: Socialization (38.12% variance)* | | | | α = 0.92 | α = 0.94 |
| Q1. I stay away from other people | 1.87 (1.30) | 0.08 | -1.04 | 0.66 | 0.58 |
| Q2. I love meeting new people (R) | 1.74 (1.30) | 0.18 | -1.02 | 0.58 | 0.74 |
| Q3. People bother me | 1.54 (1.31) | 0.42 | -0.91 | 0.73 | 0.68 |
| Q4. I feel uncomfortable around other people | 1.36 (1.32) | 0.61 | -0.76 | 0.79 | 0.71 |
| Q5. I do not like to be seen by others | 1.15 (1.24) | 0.83 | -0.40 | 0.76 | 0.59 |
| Q6. It is hard for me to join in on groups | 1.32 (1.35) | 0.71 | -0.70 | 0.66 | 0.74 |
| Q7. I enjoy being in a social situations (R) | 1.56 (1.24) | 0.35 | -0.85 | 0.69 | 0.76 |
| Q8. I avoid talking with other people | 1.22 (1.22) | 0.71 | -0.42 | 0.78 | 0.73 |
| Q9. I much prefer to be alone than with others | 1.61 (1.37) | 0.38 | -1.03 | 0.81 | 0.68 |
| Q10. I strongly prefer to be around other people (R) | 1.46 (1.25) | 0.46 | -0.80 | 0.79 | 0.74 |
| Q11. I do not enjoy social interactions | 1.31 (1.23) | 0.62 | -0.57 | 0.75 | 0.64 |
| *Factor 2: Isolation (9.61% variance)* | | | | α = 0.84 | α = 0.88 |
| Q12. I spend most of my time at home | 2.36 (1.27) | -0.30 | -0.92 | 0.46 | 0.68 |
| Q13. I shut myself in my room | 1.15 (1.34) | 0.84 | -0.54 | 0.55 | 0.50 |
| Q14. I spend most of my time alone | 1.33 (1.35) | 0.62 | -0.87 | 0.64 | 0.73 |
| Q15. I rarely meet people in-person | 1.15 (1.22) | 0.78 | -0.43 | 0.66 | 0.63 |
| Q16. I do not live by society's rules and values | 1.28 (1.11) | 0.52 | -0.59 | 0.62 | 0.45 |
| Q17. I have little contact with other people talking, writing, and so on | 1.20 (1.19) | 0.67 | -0.49 | 0.80 | 0.51 |
| Q18. I rarely spend time alone | 1.50 (1.06) | 0.51 | -0.21 | 0.62 | 0.54 |
| Q19. I spend very little time interacting with other people | 1.67 (1.19) | 0.20 | -0.79 | 0.80 | 0.55 |
| *Factor 3: Emotional support (5.14% variance)* | | | | α = 0.73 | α = 0.89 |
| Q20. There really is not anyone with whom I can discuss matters of importance | 1.80 (1.01) | -0.19 | -0.99 | 0.43 | 0.76 |
| Q21. There are people in my life who try to understand me (R) | 1.55 (1.11) | 0.31 | -0.55 | 0.61 | 0.63 |
| Q22. I can share my personal thoughts with several people (R) | 1.63 (1.16) | 0.32 | -0.71 | 0.61 | 0.63 |
| Q23. There are few people I can discuss important issues with | 1.65 (1.21) | 0.12 | -1.11 | 0.69 | 0.62 |
| Q24. There is really no one very significant in my life | 1.79 (1.29) | 0.16 | -1.14 | 0.62 | 0.55 |
| Q25. I have someone I can trust with my problems (R) | 1.84 (1.06) | 0.38 | -0.30 | 0.71 | 0.85 |

(R), all items were rated on a scale ranging from 0 (strongly disagree) to 4 (strongly agree); α, Cronbach's alpha;

^, results are from Teo et al. [50].

0.92. Cronbach's alpha for the subscales was 0.92 (socialisation, 11 items), 0.84 (isolation, 8 items) and 0.73 (emotional support, 6 items). A detailed description of the factor loadings on each item, the internal consistency of the current study, and the findings of Teo et al. [50] are presented in Table 2.

### 3.3. Confirmatory factor analysis

CFA was carried out with 765 participants on a three-factor model with 25 items. Table 3 and Fig 1 demonstrate CFA in standard conditions with a coefficient (Fig 1). As the results showed, none of the items was removed from the subfactor, as its factor loading was higher than the critical value. As listed in Table 3, the correlation coefficients between HQ-25 and its subscales were positive and significant in all cases. Covariance among the three-factor model on socialisation and Isolation (0.77, z = 113.60, p < .001), Socialization, and emotional support (0.58, z = 3.26, p = 0.001) and Isolation and emotional support (0.43, z = 2.88, p = 0.004). Furthermore, given the fit indices ($\chi^2/df$ = 17.92, p < .001; CFI = 0.90; TLI = 0.95; RMSEA = 0.04, 95% CI 0.02–0.07; SRMR = 0.05) of the three-factor model, it has an acceptable goodness of fit and fits the collected data. Eventually, based on EFA and CFA, the Omani version of HQ-25 was confirmed for Omani society with 25 items and three sub-factors.

**Table 3. Estimated loading, residual variances, and factor covariance on the Arabic version of HQ-25 using confirmatory factor analysis.**

| Factor | Question | Loading (95% CI) | Residual variances (95% CI) |
|---|---|---|---|
| **1: Socialization** | 1 | 0.80 (0.72–0.88)* | 0.94 (0.84–1.04)* |
| | 2 | 0.62 (0.53–0.71)* | 1.39 (1.25–1.53)* |
| | 3 | 0.92 (0.83–0.99)* | 0.89 (0.80–0.99)* |
| | 4 | 0.99 (0.92–1.07)* | 0.65 (0.58–0.73)* |
| | 5 | 0.96 (0.88–1.03)* | 0.73 (0.65–0.81)* |
| | 6 | 0.92 (0.84–1.01)* | 0.86 (0.77–0.96)* |
| | 7 | 0.83 (0.74–0.91)* | 0.98 (0.88–1.09)* |
| | 8 | 0.95 (0.88–1.03)* | 0.64 (0.57–0.71)* |
| | 9 | 1.08 (0.99–1.16)* | 0.71 (0.63–0.79)* |
| | 10 | 0.88 (0.79–0.96)* | 0.87 (0.77–0.96)* |
| | 11 | 0.92 (0.84–0.99)* | 0.76 (0.68–0.84)* |
| **2: Isolation** | 12 | 0.64 (0.55–0.73)* | 1.23 (1.10–1.36)* |
| | 13 | 0.95 (0.86–1.04)* | 0.97 (0.87–1.08)* |
| | 14 | 1.10 (1.02–1.18)* | 0.63 (0.55–0.71)* |
| | 15 | 0.83 (0.75–0.91)* | 0.88 (0.78–0.97)* |
| | 16 | 0.41 (0.33–0.49)* | 1.05 (0.95–1.16)* |
| | 17 | 0.76 (0.68–0.83)* | 0.80 (0.72–0.89)* |
| | 18 | 0.18 (0.11–0.26)* | 0.97 (0.87–1.07)* |
| | 19 | 0.83 (0.75–0.90)* | 0.77 (0.69–0.86)* |
| **3: Emotional support** | 20 | 0.68 (0.60–0.76)* | 0.64 (0.55–0.72)* |
| | 21 | 0.48 (0.38–0.57)* | 1.28 (1.14–1.41)* |
| | 22 | 0.36 (0.27–0.46)* | 1.22 (1.09–1.34)* |
| | 23 | 0.41 (0.31–0.50)* | 1.39 (1.24–1.53)* |
| | 24 | 0.98 (0.88–1.07)* | 0.79 (0.67–0.91)* |
| | 25 | 0.85 (0.77–0.92)* | 0.49 (0.41–0.58)* |

95% CI, 95% confidence interval;

*, sig. at p<0.001;

Factor covariance, Isolation of socialisation: 0.77 (95%CI 0.61–0.89), Emotional support: 0.58 (95% CI 0.40–0.67), Emotional support: 0.43 (95% CI 0.33–0.59).

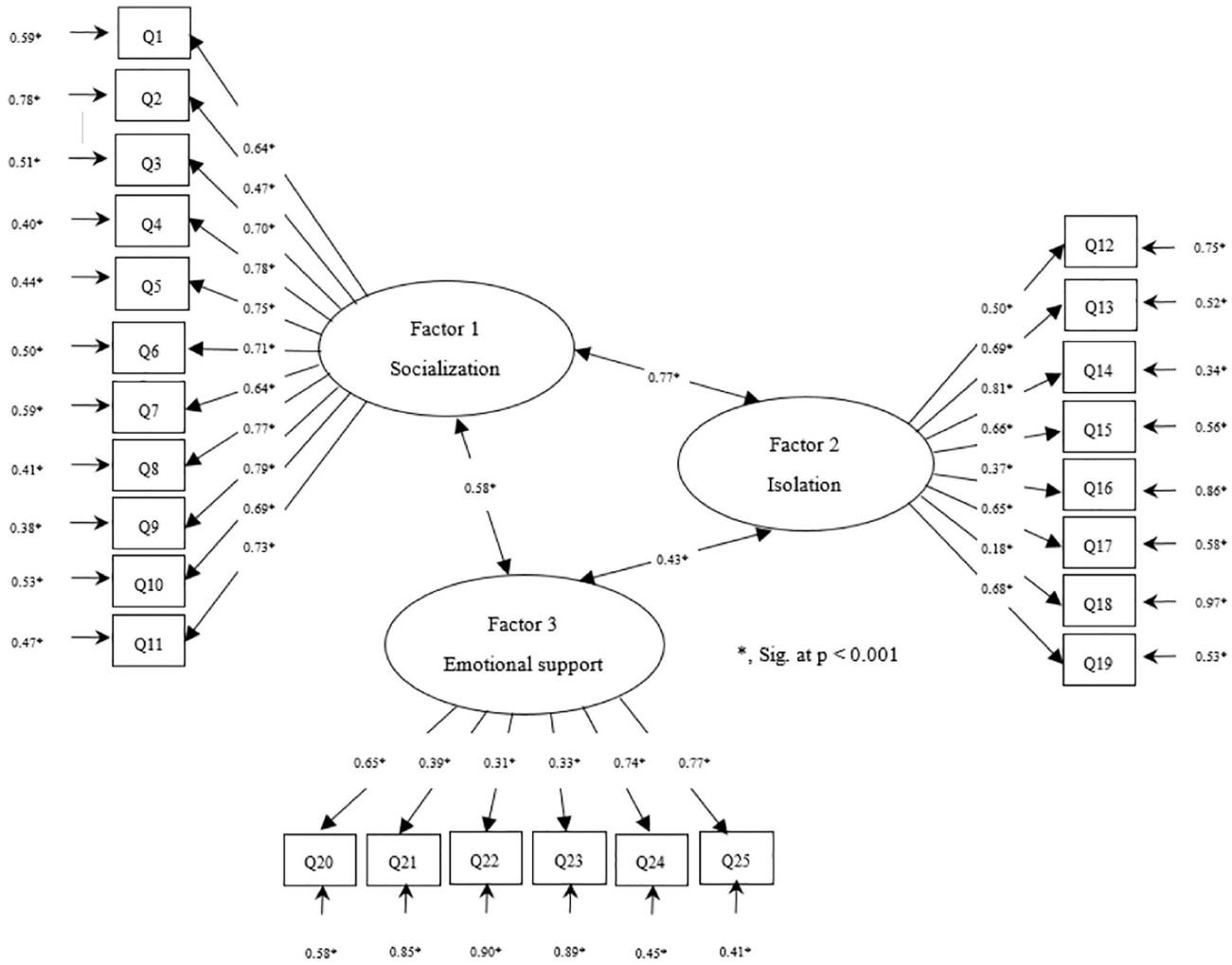

**Fig 1. Confirmatory factor analysis depicting the three-factor model for the Hikikomori Questionnaire (HQ-25).**

## 3.4. Risk factors associated with the presence of HLID

Table 4 shows the univariate and logistic analysis of sociodemographic and clinical outcomes associated with subjects who had HLID compared to those who did not. In the univariate analysis, results showed that female gender (HLID vs. non-HLID: 71.0% vs. 65.1%, p = 0.014), non-married status (52.2% vs. 27.9%, p<0.001), age (p<0.001), place of living (35.3% vs. 40.6%, p = 0.001), living alone (24.8% vs. 10.2%, p<0.001), being a student (25.0% vs. 10.2%, p<0.001) or unemployed (30.2% vs. 24.9%, p<0.001), and screen time of more than 7 hours (26.4% vs. 8.1%, p<0.001), or 4–7 hours (44.0% vs. 33.8%, p<0.001) were shown to be significantly associated with the present of HLID.

Among clinical outcomes, having a mental illness (12.6% vs 3.6%, p<0.001), a family history of mental illness (23.5% vs. 16.9%, p = 0.002) and adverse childhood experiences (ACE) (73.7% vs. 48.7%, p<0.001) showed a significant association with the presence of HLID. Factors in univariate logistics are significant at 5% and will be further analysed using multivariate logistic analysis (enter method) analysis. The results showed that single/divorced/widowed marital status (p = 0.007), age (p = 0.008), place of living (p = 0.006), unemployed occupational

**Table 4. Univariate and logistic regression analysis in patients with the Hikikomori-like idiom of distress (HLID)according to HQ-25 associated with demographic and clinical outcomes.**

| | The presence of HLID [a] | | Logistic regression | | |
| --- | --- | --- | --- | --- | --- |
| | Yes (n = 673) | No (n = 856) | Univariate | Multivariate | |
| Outcomes | n (%) | n (%) | OR (*p*-value) | OR (*p*-value) | VIF |
| *Demographic* | | | | | |
| Gender: | | | | | |
| Female | 478 (71.0) | 557 (65.1) | 1.32 (0.014*) | 1.09 (0.528) | 1.107 |
| Male | 195 (29.0) | 299 (34.9) | | | |
| Marital status: | | | | | |
| Single/divorced/widow | 351 (52.2) | 239 (27.9) | 2.81 (< .001**) | 1.51'(0.007*) | 1.714 |
| Married | 322 (47.8) | 617 (72.1) | | | |
| Age (years) | | | 0.93 (< .001**) | 0.97 (0.008**) | 2.017 |
| Education level: | | | | | |
| High school/below | 102 (15.2) | 119 (13.9) | 1.11 (0.489) | | |
| University/above | 571 (84.8) | 737 (86.1) | | | |
| Place of living: | | | | | |
| Urban | 224 (35.3) | 356 (40.6) | 0.71 (0.001**) | 0.71 (0.006**) | 1.010 |
| Rural | 449 (64.7) | 500 (59.4) | | | |
| Socioeconomic status: | | | | | |
| Upper | 34 (5.1) | 28 (3.3) | 1.51 (0.117) | | |
| Middle | 235 (34.9) | 325 (38.0) | 0.90 (0.333) | | |
| Low (ref) | 404 (60.0) | 503 (58.8) | | | |
| Living alone: | | | | | |
| Yes | 167 (24.8) | 87 (10.2) | 2.92 (< .001**) | 0.39 (0.3361) | 3.395 |
| No | 516 (75.2) | 769 (89.8) | | | |
| Occupation: | | | | | |
| Student | 168 (25.0) | 87 (10.2) | 3.56 (< .001**) | 3.34 (0.240) | 3.242 |
| Unemployed | 203 (30.2) | 213 (24.9) | 1.75 (< .001**) | 1.72 (< .001**) | |
| Employed (ref) | 302 (44.9) | 556 (65.0) | | | |
| Screen time spent on the Internet: | | | | | |
| >7 | 178 (26.4) | 69 (8.1) | 6.45 (< .001**) | 4.50 (< .001**) | 1.168 |
| 4–7 | 296 (44.0) | 289 (33.8) | 2.56 (< .001**) | 2.08 (< .001**) | |
| <4 (ref) | 199 (29.6) | 498 (58.2) | | | |
| Having mental illness: | | | | | |
| Yes | 85 (12.6) | 31 (3.6) | 3.85 (< .001**) | 3.68 (< .001**) | 1.042 |
| No | 588 (87.4) | 825 (96.4) | | | |
| Family mental illness: | | | | | |
| Yes | 158 (23.5) | 145 (16.9) | 1.50 (0.002**) | 1.18 (0.260) | 1.069 |
| No | 515 (76.5) | 711 (83.1) | | | |
| Adverse Childhood Experiences^: | | | | | |
| Yes | 496 (73.7) | 417 (48.7) | 2.95 (< .001**) | 2.59 (< .001**) | 1.085 |
| No | 177 (26.3) | 439 (51.3) | | | |
| Smoking: | | | | | |
| Yes | 39 (5.8) | 37 (4.3) | 1.36 (0.190) | | |
| No | 634 (94.2) | 819 (95.7) | | | |
| Drinking alcohol: | | | | | |
| Yes | 12 (1.8) | 10 (1.2) | 1.54 (0.320) | | |

(*Continued*)

**Table 4.** (Continued)

| | The presence of HLID [a] | | Logistic regression | | |
|---|---|---|---|---|---|
| | Yes (n = 673) | No (n = 856) | Univariate | Multivariate | |
| Outcomes | n (%) | n (%) | OR (*p*-value) | OR (*p*-value) | VIF |
| No | 661 (98.2) | 846 (98.8) | | | |

[a]HQ-25: *Hikikomori Questionnaire*, total score range 0–100, cut-off is 42 (Yes) and <42 (No);

^: Physical/emotional/sexual abuse;

Logistic (enter) model: Hosmer and Lemeshow ($\chi^2$ = 3.30, p = 0.914), Cox and Snell $R^2$ = 0.204, Sensitivity = 66.4%, specificity = 73.0%, overall = 70.1%; Ref: reference point; a: OR: Odds ratio;

*, Sig. at *p*<0.05;

**, Sig. at *p*< 0.01;

VIF: Variance inflation factor, cut-off > 5 considered problematic.

status (p<0.001), screen time beyond 4–7 hours (p<0.001) or >7 hrs (p < .001), comorbid mental illness (p<0.001), and ACE's (p<0.001) were significant risk factors for HLID. According to the Hosmer and Lemeshow goodness of fit test ($\chi^2$ = 3.30, p = 0.914), the model has an adequate prediction power of 70.1% with Cox and Snell $R^2$ of 0.20. The multicollinearity of the risk factor in the logistic regression analysis ranged from 1.042 to 3.395, indicating an acceptable level. Single/divorced/widow participants were 1.5 times (OR = 1.51, 95% CI 1.12 to 2.03, p = 0.007) more likely to have HLID than married individuals. Older participants were less likely (OR = 0.97, 95% CI 0.95–0.99, p = 0.008) to have HLID than younger participants. Those who lived in the urban area were 0.71 times (OR = 0.71, 95% CI 0.56 to 0.91, p = 0.006) less likely to have HLID than those who lived in the rural area. Unemployed participants were 1.7 times (OR = 1.72, 95% CI 1.30 to 2.28, p < .001) more likely to have HLID than those employed. Those who spent more than 7 hours and 4–7 hours per day on the Internet were 4.5 times (OR = 4.50, 95% CI 3.16 to 6.41, p < .001) and 2.1 times (OR = 2.1, 95% CI 1.61 to 2.70, p < .001), respectively, more likely to have HLID than those who spent less than 4 hours on the screen per day. Among the clinical risk factors, those with a history of mental illness were 3.7 times (OR = 3.7, 95% CI 2.29 to 5.91, p < .001) more likely to have HLID than those without a history of mental illness. Those who had experienced ACE were 2.6 times (OR = 2.6, 95% CI 2.03 to 3.20, p < .001) more likely to have HLID than those without such experiences.

## 4 Discussion

The social behaviour of the average Omani citizen is speculated to have been affected by the consequences of the rapid economic development and modernisation of the country [45]. Along with higher standards of living and urbanization, the country has witnessed increased healthcare utilization for problems related to maladjustment similar to what has been documented in other similar societies as well, although some are experienced in a more socio-cultural context and reflect characteristic local idioms of distress [9]. In the collectivist vs. individualist social dichotomy, traditional Omani society falls under the former [66]. The growing corpus of literature has suggested that the introduction of features of postmodernism into traditionally collective societies has the potential to trigger maladjustment due to their antithetical characteristics, including pathological social withdrawal and HLID [40, 67]. There is a paucity of literature on the impact of imposed social withdrawal due to the COVID-19 pandemic. Therefore, this study was initiated to address three interrelated goals. The first aim was to examine the factorial validity of the 25-item *Hikikomori Questionnaire* (HQ-25). The

second objective is to examine the prevalence and the third is to explore factors associated with HLID after lifting the lifting of SARS-COV-2 in the Arabian Gulf country of Oman.

Despite the wide coverage of HLID in the popular media and scientific literature, few studies examine the epidemiological prevalence of HLID and comparisons between these studies are hampered by the fact that different studies used divergent catchment areas, screening tools and different definitions of HLID. Therefore, the present study had embarked to address two interrelated objectives. The first was to examine whether established HLID phenotypes have heuristic value in the Omani population. For this, exploratory factor analysis (EFA) and confirmatory factor analysis (CFA) methods have been analysed to confirm the factorial validity of the *Hikikomori Questionnaire* (HQ-25) [50, 51]. One of the most pervasive issues discussed in behavioural and human sciences is the tendency to indiscriminately adapt amorphous terms and entities, such as 'poor coping', to different cultural and ethnic groups [68]. This view has been poignantly articulated by Kleinman [69] as 'the reification of one culture's diagnostic categories in one culture and their projection onto patients in another culture, where those categories lack coherence and their validity has not been established' (p. 14). To avoid this categorical fallacy, it is essential to develop diagnostic measures derived from local culturally sensitive idioms of distress. However, if different measures are developed, they have the potential to hinder the comparison of results between international respondents. One way to bypass these issues is to adopt instruments that have been developed from the data of one culture to another. Although translating into another language has clear heuristic value, it could still be hindered or misinterpreted by subtle linguistic and conceptual errors. Therefore, this study has also used exploratory factor analysis (EFA) to determine whether responses to questions could be grouped into scales that measure a common factor. Teo et al. [50] developed the 25-item *Hikikomori Questionnaire* with a three-factor structure and it has been used in several international epidemiological surveys [32, 50, 70]. The factorial validity, EFA and CFA were carried out on the 1529 participants. The EFA results showed that 50.08% of the variance of the 25 items was attributed to the original three-factor model. This echoes the initial three-factor structure of Teo and Gaw [8]. The present three-factor model is also supported by a study conducted among Italian [51] and Chinese [32] respondents, furthering the proposed hypothesis that HLID might be best described as a culture-reactive idiom of distress rather than as a culture-bound one.

The second aim was to explore the prevalence of HLID in the Omani community. The results of these two aims are recapitulated below in the context of existing international literature in tandem. Using the adopted Arabic version of HQ-25, the present study suggested that 44% of the participants (n = 673) exhibited HLID in Oman. Similar studies in other populations have reported that the prevalence of HLID typically falls within the range of 0.87% to 26.7% [2, 12, 31, 71, 72]. In a community survey in China, using HQ-25 among a sample of 1,066 youth, approximately 8.1% of the cohort were 'marked with social isolation at home or withdrawal with a duration of at least 3 months" and "marked with social isolation in one's home and withdrawal with a duration of at least 3 months' [32]. In Hong Kong, Wong et al. [12] have reported prevalence rates of 1.9% for social HLID for 6 months or more, 2.5% for 6 months or less, and 2.6% prevalence of self-perceived non-problematic social withdrawal. In Japan, where HLID first came to prominence, Koyama et al. [71] reported a prevalence rate of HLID in the cohort of a national face-to-face household survey. Trends in the literature suggest that the prevalence rate of HLID is higher side when requested via self-rated online surveys [32, 41] and that these figures become more conservative when using face-to-face interviews, such as those that adhere to semi-structured standardised interviews. This pattern is consistent with what is often reported in psychiatric epidemiological studies [73]. The question that arises is whether this has resulted in a prevalence rate in the present study that

appears to exceed previously reported frequencies from other countries. It is worth noting that this study was conducted using the Arabic version of HQ-25 and a series of confirmatory factors were performed, with the results supporting the same factor structures as the original version of HQ-25 [50]. One possibility is that the presently used cutoff score of 42 (out of 100) is the same as that proposed by the creators of the HQ-25 during its development with a Japanese sample. The present study did not go so far as to assess the diagnostic accuracy of the psychometric properties with the present Omani sample, which would have required gold-standard diagnostic interviews and might have yielded a different cut-off score. Teo et al. [50] reported accruing a sensitivity of 94% and specificity of 61% via receiver–operator curve (ROC) among clinical and community respondents in southern Japan, during the development of the questionnaire. Similarly, among the Chinese population, Hu et al. [32] reported that the cutoff of 42 was associated with 93.48% and 90.74% sensitivity and specificity, respectively. The second possibility is that, while this study has a reasonably high response rate of 76.5%, it is nonetheless a highly focused study with a very specific outcome measure. In a similarly focused online study in Oman examining the prevalence of social anxiety disorder, which shares several features with HLID, the prevalence rate was also inclined towards the higher side compared to current international trends [40]. In this study, it was speculated that shy individuals, due to their temperament and tendency to report greater Internet and screen time, may be more motivated to respond and participate in online surveys. It can be further suggested that such individuals may find that the concepts and items of the HQ-25 are salient to themselves and are capable of capturing their 'inner world', causing them to be more enthusiastic to participate in this survey [74]. More studies are needed to examine the theoretical value of such a confounder.

Third, assuming that HQ-25 did not provide spurious results, one may wonder whether the population from which the sample is derived adequately utilises existing biomedical, mental health, and social services. Although Oman has developed a universal free healthcare system for behavioural and mental health problems, the use of such services leaves much to be desired [75]. In traditional Omani society, behavioural and mental problems, as described in biomedical parlance, are often considered the prerogative of traditional healing systems, rather than relatively new but scientifically proven methods and services [75]. Associated with this, studies have suggested that the pathways to care are considerably modified by local, social and psychological characteristics [76]. With the family-centred nature of Omani society [77], the family plays a critical role in defining the care pathways of an individual. There is also evidence to suggest that those who exhibit externalized behavioural problems (physical aggression, verbal bullying, relational aggression, defiance, theft, and vandalism) are more likely to seek consultation from biomedical care when compared to those with internalized behavioural problems (withdrawal, marked with social avoidance or overt social phobia) [49, 75]. It could be hypothesized here that individuals with externalized behavioural problems are likely to be perceived as 'troublemakers', and hence bring upon societal shame and disruptions to family functioning. On the contrary, people with avoidance and social withdrawal-like behaviour often feel comfortable with their 'escapism', as this behaviour is generally syntonic with the expression of their distress, while also being ignored and sometimes encouraged by families in conservative societies. Therefore, they are more likely to remain in their community without the motivation to access biomedical care [75].

Finally, it should be noted that the present study was conducted during the SARS-COV-2 pandemic. In Oman, reflecting the rest of the world, a significant proportion of the population contracted SARS-COV-2 [78], and the government introduced physical and social distancing, quarantine methods, and a curfew to minimise the risk of infection transmission. Although the current data was collected at a time when these government-imposed social restrictions

were largely lifted, the fallout from the previous year was increasingly coming to light at this time. Such measures resulted in curtailing people from going to work and school which, in turn, led to the proliferation of 'telework' and online learning. The local economy and labour force were severely dented, and thus, options for aspirant job-seekers to secure meaningful employment became untenable in a country with an increasingly high number of job-seekers even prior to the onset of the pandemic [79]. There is also evidence to suggest that while the country ceased to accommodate much-needed employment in government sectors, emerging private sectors catering to the tourism and services industry are perceived to be a less attractive alternative for career aspirations of Omanis [80]. In a society where education, occupation and 'isms and schisms' are of great significance, some vulnerable young people who do not live up to societal expectations may be more vulnerable to developing symptoms of HLID. It is not clear whether the SARS-COV-2 pandemic and its subsequent tribulations have further exacerbated maladjustment among these groups of individuals with unstable employment. Studies from several countries have reported an increase in poor mental health outcomes during the SARS-COV-2 pandemic, including increasing reports of those with HLID [27, 81, 82] as well as maladjusted temperaments and poor coping strategies that are commonly hypothesised to be precursors closely related to HLID [83].

The third related objective of the present study was to examine the factors associated with the development of HLID. In the present study, age, marital status, living alone, occupation, screen time, having a mental illness or a significant family history of mental illness and those with risk ACE were significant factors associated with HLID in the regression analysis. These are recapitulated below in tandem with the extant literature and sociocultural teaching of Oman.

The mean age for the present cohort was 31.2±7.7 with a range of 19.0 to 45.0. In the regression analysis, younger adults were more likely to have manifested HLID than older participants. Previous studies have focused on the assumption that HLID is a challenge that often affects young people, with a starting age ranging from 7 to 22 years. Rather than being a predominant youth challenge, there is evidence to suggest that HLID is also found in the adult population. Malagón-Amor et al. [17], in their descriptive study of the Spanish population, have reported that the mean age at the onset of HLID was 40 years old, with a mean socially withdrawn period of 3 years. In other parts of the world, social isolation or 'loneliness' is often equated with the elderly and frail [84]. Therefore, HLID can be expressed in a complex way according to varying sociocultural factors. For this reason, this study has the advantage of exploring the general population. With future studies and more data, the assumption that HLID is typically expressed primarily by young people may change. In terms of marital status, most of the sample was married (61.4%). However, in the regression analysis, when marital status was grouped as married or otherwise (single, divorced, or widowed), being unmarried increased the probability of exhibiting HLID. This is consistent with the view that being in a married relationship hinders one's tendency towards social withdrawal. It is possible that those who show HLID and other signs of social withdrawal also avoid marriage. There is also the possibility that marriage and social isolation might be associated with age. The average age of marriage in Oman has been documented to be 20 years, with the median age of the first marriage being 16 years according to the Oman National Health Survey [85]. Recent data suggest that the preferred age for marriage falls within the age range [85]. In a rural population residing in Akita Prefecture, northern Japan, Yong et al. [86] reported that among those who had HLID, 57.3% were married. Therefore, in terms of marital status, the situation in Oman does not mirror the trends observed in Japan. Certain unique characteristics common to Arab families, such as the traditional tribal organisation of families within Omani society, may offer insight into the role of family and marital dynamics in the development of maladaptive coping

and HLID [87]. This implies that when comparing the characteristics of HLID with other studies, it is important to consider the definition of hikikomori used in each study. The example, Yong et al. [86], almost half of the people labelled as "hikikomori" were full-time workers and many were married, which differs from the typical definition of hikikomori as someone who is not working. Therefore, it is important to take into account the definition of hikikomori used in each study to accurately compare their findings. Furthermore, the high proportion of married individuals in the study by Yong et al. [86] may reflect a difference in definitions rather than a characteristic specific to hikikomori. With contradictory findings on the role of marital status in the development of HLID, future studies exploring its effects on the development of HLID should consider what factors culture has to play on marital dynamics.

Oman has also begun to witness a rapid decline in extended multigenerational families living in the same household and a growing number of nuclear families [87]. It is also worth considering that while early marriage used to be a common pattern in Oman, the average age upon marriage has been incrementally rising, particularly for women who typically reside within their parents' household until marriage [88]. Within this background, this study indicates that approximately 25% of the participants who lived alone also endorsed the presence of HLID, and this was proven to be statistically significant in the univariate analysis. Oman is an emerging economy that is currently going through the second phase of demographic transition when society tends to have a high birth rate and a boom in young people [85]. However, it is possible that society, due to rapid modernisation fostered by high income, may also be at a crossroads with the third phase of demographic transition, particularly in some segments of urban society. In the third phase of demography in transition, high birth rates tend to decline, the population grows at a lower rate, and traditional interdependence also declines, which has been increasingly observed in Omani society [89]. Another feature of the third phase of demographic transition is the consolidation of competitive capitalist-orientated social expectations [12, 13], which, in turn, has caused the rise of 'anomie' where, as Durkheim describes, there is dysfunction between personal or group standards and wider social standards [73, 90]. Such a mismatch could be a salient factor that results in an increase in the number of people living in nuclear families or alone. In studies that have examined factors associated with HLID, loneliness has been frequently cited [32]. A study by Malagón-Amor et al. [17] also points to the positive and negative influence of family presence on HLID, but it should be noted that participants that lived with their families reported that family members were more likely to have perceived their social withdrawal first and showed greater cooperation in initiating and formulating their treatment plans.

Social withdrawal would imply that the affected individual is incapable of meaningful common forms of employment. Since the present instrument (HQ-25) takes advantage of the subclinical variation of HLID, it might be worthwhile to explore this among the general population. In the present study, the occupation of the cohort constituted a 16.7% student population, 27.2% job seekers or unemployed, while the rest of the majority (56.1%) were employed. Among these, 25% of students, 30.2% of the unemployed, and 44.9% of the employed met the subclinical HLID threshold. In the regression analysis, being a student or unemployed was found to be significant. There is a dearth of studies on the relationship between HLID and employability. In an ideographical study, Iwakabe [91] reported that some people with HLID tend to gain employment but are often prone to quit and that quitting tends to herald reclusive isolation. It is also important to reiterate that this study occurred a year after the SARS-COV-2 outbreak, when 'working from home' and remote work became more commonplace and, at times, even encouraged after the lifting of social restrictions. The resulting isolation resulted in decreased productivity and increased job insecurity [92]. More studies are needed to examine the quality of life and the types of meaningful existence pursued by people with HLID.

Oman has been reported to have 3.3 million internet users, which amounts to 71% of the population. Increasing standards of living have left its population with high literacy levels (>97%) which, in turn, would imply that Omani society is likely to have a large number of users of online social media [93]. Some studies have alluded to the view that hikikomori-like temperaments have a strong temporal relationship with internet addiction or high screen time [32, 70, 74, 93]. This is consistent with the present cohort, whose screen time was high and had a significant temporal relationship with the presence of HLID in logistic regression. With HLID and its development being a relatively new subject in the scientific literature, it has not yet been established whether internet addiction is a contributing factor to social withdrawal and HLID, or whether HLID increases susceptibility to internet addiction [74].

Parallel to the high presence of high screen time, there is a growing corpus of literature that suggests that *hikikomori* falls into two subtypes [94]. One is the 'pure' idiopathic type, or primary hikikomori, while secondary hikikomori is comorbid with other psychiatric conditions. According to Pozza et al. [95], secondary hikikomori tends to have a prodromal phase of psychosis, depressive disorders and related mood disorders, social anxiety disorder, post-traumatic stress disorder and related disorders, obsessive-compulsive disorder and related disorders, generalised anxiety disorder, panic disorder/anxiety, internet addiction/game, obsessive-compulsive personality disorder, dependent personality disorder, and autism spectrum disorders. The diagnosis of secondary hikikomori has been suggested to be more socially acceptable than other established conditions, as it avoids the stigma associated with these psychiatric disorders [8]. However, there is also a dissenting view that the symptoms of social maladjustment that are intimately related to *hikikomori* have the potential to act as a 'gateway disorder', leading to more serious psychiatric conditions over time [67] (p. 1052). In addition to the primary vs. secondary peculiarity, some studies have also suggested that HLID could be considered as a spectrum of 'internet addiction' [96] or a veneer of 'modern-type depression' [30]. The present study suggests that both primary and secondary HLID exist in Oman. 12.6% of the HLID sample reported co-morbid distresses, such as depression, anxiety, social phobia, obsessive-compulsive disorder, and personality disorder. However, 87.4% of this appears to have primary HLID, if self-reported results can be taken at face value. This majority of primary HLID may also be associated with the fact that it is more refractory to treatment [54]. Therefore, there is merit in exploring co-morbidities or otherwise using a more robust methodology.

The final variable found to be associated with HLID was factors associated with the formative years of the participants. Adverse childhood experiences (ACEs) include physical, verbal/emotional, and sexual adverse experiences. The term also includes neglect and growing up in a dysfunctional family. For the present study, participants were simply asked if they could recall having had an ACE. Among those with subclinical HLID, 73.7% endorsed having a history of ACE. The present data are consistent with other nascent literature documenting the association between a history of trauma, peer rejection, or dysfunctional family dynamics and the development of HLID [4, 41, 54, 97, 98]. Socio-cultural constraints and traditional belief systems in Oman have been suggested to put children at risk, and although data is sparse, studies indicate that Omani society is not immune to child abuse and neglect [66, 70]. The types of ACE and their relationship to the development of HLID is an important topic that needs to be further explored in future studies.

## 4.1. Limitations

Some of the obvious limitations of this study are worth highlighting. <u>First</u>, in previous studies, diagnostic criteria for *hikikomori* as active symptoms lasting for 3 months vs. 6 months of persistent social isolation and withdrawal have both been considered. This is one of the factors

that limit the generalization of the present study, since the present subclinical scale, the *Hikikomori Questionnaire-25*, was not equipped to capture the duration of HLID. Second, given the absence of a cut-off point for the Omani population, it is crucial to approach the interpretation of the significant risk factors identified in this study with caution. Some of the clinical risk factors were asked via a self-reported questionnaire developed by the authors of the present study. For example, psychiatric comorbidities would likely have been more valid and less inflated had they been explored using established structured self-reported scales. The lack of use of a structured interview may also explain why ACE was highly reported (59.7%) in the overall sample. However, such an undertaking would have made the study survey excessively long to complete, which, in turn, would have reduced the response rate. Therefore, to address this in subsequent studies, community surveys would be required using validated questionnaires. Third, the cutoff score of 42 (out of 100) utilised is the same as that proposed by the creators of the HQ-25 during its development [50] and follow-up studies [32]. The current study did not go as far as to examine the factorial validity of HQ-25 by including external criteria such as the UCLA loneliness scale [99] and assess the diagnostic accuracy of the psychometric properties of HQ-25 of HQ-25 with the present Omani sample, which would have required diagnostic interviews with the gold stand and could have produced a different cut-off point. There is merit in examining the sensitivity, specificity, and predictability of HQ-25 in different populations so that culturally specific cut-off points can eventually be established. Furthermore, this study did not collect the exact frequency or duration of the decrease in social participation (that is, the exit patterns of the participants and socialising), which may also have affected the results. Fourth, this study was conducted a year after the official declaration of the SARS-Cov-2 pandemic in March 2021. While governmental social restriction measures were mostly lifted at the time of this study [43], the population's fears or sense of civic duty towards preventing infection spread might have been a possible confounder with regard to the prevalence of HLID. For this study, this has been considered as a form of self-imposed social withdrawal adjacent to the concept of *hikikomori*, and the potential impacts of previous restrictions have been theorised and discussed. It is important to collect data during adverse events that are likely to exacerbate the increasing rates of HLID, allowing the following studies to build on and further contextualise these results. Fifth, this study included three tasks: exploring the prevalence of HLID, the validity of HQ-25 in a local context, and determining the associated risk factors with HLID in the general population. The respondents required for each task should be ideally calculated separately. Sixth, this study has used snowball techniques. It was therefore equipped to encourage recipients to cascade the link to others in their network can also increase the reach of the survey. However, there are likely to be two sources of attrition. The presently used recruitment approach may not be representative of the entire population, as it relies on individuals who are active on social media and willing to participate in the survey. Secondly, there might be biases among the respondents due to the snowball technique used, which is a non-probability sampling method. This means that individuals who are more socially connected or have a higher interest in the topic might be more inclined to participate and share the link. Consequently, these factors could potentially influence the results, particularly the construct of HQ-25 in the CFA. Within this background, that online surveys are increasingly used to tap into trends in the community, it remains to be seen whether snowball techniques are superior to accruing more response rate or whether there are sociocultural factors that shape response rate. Therefore, empirical scrutiny of online recruitment is warranted. Finally, the cross-sectional study design used in the current study was not equipped to unravel the 'cause and effect'. Thus, the interpretation of risk factors for HLID should be cautious and a case-control study based on the present findings should be highly recommended for future studies.

## 5 Conclusions

The discourse on *hikikomori* being a cultural reactive phenomenon rather than a culturally specific one has recently regained prominence with the growing emergency of social withdrawal, loneliness, and, for the present study, the Hikikomori-like idiom of distress (HLID). This study examines whether self-imposed social withdrawal, coinciding with the SARS-COV-2 pandemic and its aftermath, has the potential to trigger HLID. The present study found that 44% of the respondents have HLID. In terms of factors associated with HLID, the present data suggest that being unmarried, unemployed, belonging to a young age group, having longer screen time, having comorbid psychological distress, and having adverse childhood experiences were significant risk factors for HLID. These risk factors appear to be based on specific sociocultural factors of Oman, as well as being integral characteristics of collective social patterns and the vagaries of modernization and acculturation. Since HLID appears to occur in diverse populations, the present data posit the view that HLID is a culture-reactive idiom rather than a culture-specific idiom of distress.

## Author Contributions

**Conceptualization:** Nasser Al-Sibani, Samir Al-Adawi.

**Data curation:** Moon Fai Chan.

**Investigation:** Salim Al-Huseini, Nutaila Al Kharusi, Yousuf Al Hasani.

**Methodology:** Moon Fai Chan.

**Software:** Moon Fai Chan.

**Supervision:** Gilles J. Guillemin, Mohammed Al-Abri, Yousuf Al Hasani, Samir Al-Adawi.

**Writing – original draft:** Nasser Al-Sibani, Aishwarya Ganesh, Samir Al-Adawi.

**Writing – review & editing:** Salim Al-Huseini, Gilles J. Guillemin, Mohammed Al-Abri, Aishwarya Ganesh, Samir Al-Adawi.

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
