## [Decision Letter · Decision Letter 0]

12 Jan 2023

PONE-D-22-33971Exploring Hikikomori-like idiom of distress a year into the SARS-CoV-2 pandemic in Oman: factorial validity of the 25-item Hikikomori Questionnaire, prevalence and associated factorsPLOS ONE

Dear Dr. Al-Adawi,

Thank you for submitting your manuscript to PLOS ONE. After careful consideration, we feel that it has merit but does not fully meet PLOS ONE’s publication criteria as it currently stands. Therefore, we invite you to submit a revised version of the manuscript that addresses the points raised during the review process.

We look forward to receiving your revised manuscript.

Kind regards,

Supat Chupradit

Academic Editor

PLOS ONE

Manuscript Number:

Reviewers' comments:

Reviewer's Responses to Questions

**Comments to the Author**

1. Is the manuscript technically sound, and do the data support the conclusions?

Reviewer #1: Yes

Reviewer #2: Yes

2. Has the statistical analysis been performed appropriately and rigorously? 

Reviewer #1: Yes

Reviewer #2: Yes

3. Have the authors made all data underlying the findings in their manuscript fully available?

Reviewer #1: Yes

Reviewer #2: Yes

4. Is the manuscript presented in an intelligible fashion and written in standard English?

Reviewer #1: Yes

Reviewer #2: Yes

5. Review Comments to the Author

Reviewer #1: Overall, the article is interesting, and it provides an insight on the development of the Omani version of the HQ-25 with the appropriate methodology. To strengthen the article, only a few issues need to be addressed.

• There are likely to have various Hikikomori-like idioms of distress (HLID) assessment tools available to the public, but the HQ-25 was purposefully chosen. The introductory section should provide the supporting details on this matter.

• In the sampling method section, the procedure to collect the data online should be more thoroughly explained

• The discussion section seems to well describe the findings, reflecting the practical ability of the Omani-version of the HQ-25 to detect prevalence and risk factors that were able to predict HLID in the Oman sample.

Reviewer #2: 1-due to use odd ratio, thus the authors should consider about sampling size.

2-According first commnet, the auhors should examine the assumption of using odd ratio.

3-well revise and well organized article.

6. PLOS authors have the option to publish the peer review history of their article (what does this mean?). If published, this will include your full peer review and any attached files.

Reviewer #1: No

Reviewer #2: No

---

## [Author Response · Author response to Decision Letter 0]

1 Feb 2023

We would like to thank the Editor and the reviewers for their insightful comments. Please find below our responses and the recommended corrections made. 

Reviewer #1

Overall, the article is interesting, and it provides an insight on the development of the Omani version of the HQ-25 with the appropriate methodology. To strengthen the article, only a few issues need to be addressed. 

Thank you for these comments.

•There are likely to have various Hikikomori-like idioms of distress (HLID) assessment tools available to the public, but the HQ-25 was purposefully chosen. The introductory section should provide the supporting details on this matter. 

Response: This important issue has now been narrated in the text. Please refer to the section highlighted in yellow in the Introduction section. 

• In the sampling method section, the procedure to collect the data online should be more thoroughly explained

Response: Thank you. Done as suggested. Please refer to the section highlighted in yellow in the Materials and Methods section, under subheading 2.4 Sampling Method.

• The discussion section seems to well describe the findings, reflecting the practical ability of the Omani-version of the HQ-25 to detect prevalence and risk factors that were able to predict HLID in the Oman sample.

Response: Thank you for these comments

Reviewer #2: 

1-due to use odd ratio, thus the authors should consider about sampling size. 

Response: This manuscript included 3 tasks: 1) Explore the prevalence rate of HLID in the general population in Oman, 2) Validate the HQ-25 in a local context, and 3) Determine risk factors associated with HLID. We agreed that we should consider different sample sizes to address different tasks. However, we included all three results in this manuscript. So, we chose the primary one - the prevalence rate - because this is the first population-based study conducted in Oman. This could potentially be a limitation, so we narrated this in the limitations section and suggested that we should consider different samples on both validity of HQ-25 and the risk factor (odds ratio) with HLID, highlighted in yellow in the Limitations subsection of the Discussion.

2-According first commnet, the auhors should examine the assumption of using an odd ratio. 

Response: We agreed that using OR is suitable for a case-control study instead of a cross-sectional one. As we stated, this is the first population-based study conducted in Oman, and all associated risk factors with HLID in this report may be misinterpreted. We narrated this as one limitation (highlighted in yellow) and suggested that a case-control study would be highly recommended for further studies.

3-well revise and well organized article.

Response: Thank you for these comments

---

## [Decision Letter · Decision Letter 1]

2 May 2023

PONE-D-22-33971R1Exploring Hikikomori-like idiom of distress a year into the SARS-CoV-2 pandemic in Oman: factorial validity of the 25-item Hikikomori Questionnaire, prevalence and associated factorsPLOS ONE

Dear Dr. Al-Adawi,

Thank you for submitting your manuscript to PLOS ONE. After careful consideration, we feel that it has merit but does not fully meet PLOS ONE’s publication criteria as it currently stands. Therefore, we invite you to submit a revised version of the manuscript that addresses the points raised during the review process.

I apologise for the long time it has taken to get this decision to you. Unfortunately the original Academic Editor became unavailable, and as such, we invited some additional reviewers. Below, you can find the comments from several new reviewers. The reviewers have raised overlapping concerns about the statistical analysis, and we invite you to address these.

We look forward to receiving your revised manuscript.

Kind regards,

Hanna Landenmark

Staff Editor

PLOS ONE

Additional Editor Comments:

Thank you for revising your manuscript. This round 2 reviewers accept it.

Reviewers' comments:

Reviewer's Responses to Questions

**Comments to the Author**

1. If the authors have adequately addressed your comments raised in a previous round of review and you feel that this manuscript is now acceptable for publication, you may indicate that here to bypass the “Comments to the Author” section, enter your conflict of interest statement in the “Confidential to Editor” section, and submit your "Accept" recommendation.

Reviewer #1: All comments have been addressed

Reviewer #2: All comments have been addressed

Reviewer #3: (No Response)

Reviewer #4: (No Response)

Reviewer #5: (No Response)

Reviewer #6: (No Response)

2. Is the manuscript technically sound, and do the data support the conclusions?

Reviewer #1: (No Response)

Reviewer #2: Yes

Reviewer #3: Yes

Reviewer #4: Yes

Reviewer #5: No

Reviewer #6: Partly

3. Has the statistical analysis been performed appropriately and rigorously? 

Reviewer #1: (No Response)

Reviewer #2: Yes

Reviewer #3: Yes

Reviewer #4: Yes

Reviewer #5: No

Reviewer #6: No

4. Have the authors made all data underlying the findings in their manuscript fully available?

Reviewer #1: (No Response)

Reviewer #2: Yes

Reviewer #3: No

Reviewer #4: Yes

Reviewer #5: Yes

Reviewer #6: No

5. Is the manuscript presented in an intelligible fashion and written in standard English?

Reviewer #1: (No Response)

Reviewer #2: Yes

Reviewer #3: Yes

Reviewer #4: Yes

Reviewer #5: Yes

Reviewer #6: Yes

6. Review Comments to the Author

Reviewer #1: The article appears to have been revised in response to earlier suggestions; thus, the manuscript is suitable and acceptable for publication in the journal.

Reviewer #2: The authors should explain the method for EFA rotation and should show the R-sqaure statistic in the result.

very well revise

Reviewer #3: Thank you for allowing me to contribute to the review of this manuscript. I believe this study has the strength of providing some data on the characteristics of hikikomori in Oman but also has the weakness of being unclear about the definition of hikikomori and the appropriate interpretation of the analysis results. I hope these comments will help.

1. The definition of hikikomori in this study is crucial. Apart from the cutoffs of the questionnaire, hikikomori in this study should be defined. As the authors state in the Limitation, the cutoff for the HQ-25 Omani version has yet to be validated. In the Introduction, I suggest that the authors clarify the definition of hikikomori in this study and explain why they have chosen it.

2. Hikikomori is naturally associated with occupational status. Therefore, when comparing hikikomori between emerging and developed economies, the authors should also focus on differences in socioeconomic factors in those countries. I suggest the authors review previous studies and add how they recognize the association between socioeconomic factors and hikikomori.

3. This study distributed the questionnaire across multiple social media platforms. Many participants may have used more than one platform. Did the authors have procedures to prevent one participant from responding more than once?

4. The authors might want to declare no multicollinearity in logistic regression analysis.

5. In Table 2, the data from Teo et al. (2018) are probably the "Japanese" version, not the HQ-25 "English" version.

6. 3.3. Confirmatory factor analysis

The extremely high covariance (0.98) between Socialization and Isolation suggests that a two-factor or one-factor structure would be appropriate. I believe the authors should discuss whether this too-strong correlation, also reported by Amendola et al. (2022), is a problem with the HQ-25 or is specific to the Omani version. The authors might also want to provide fit index data for two-factor or one-factor models so that readers can compare them.

Amendola, S., Presaghi, F., Teo, A. R., & Cerutti, R. (2022). Psychometric properties of the Italian version of the 25-item Hikikomori Questionnaire. International journal of environmental research and public health, 19(20), 13552.

7. I agree with the authors that the lack of validity of the scale is a limitation of this study. Therefore, the risk factors results may also need to be interpreted cautiously. I think the authors need to emphasize this in Limitation. Without understanding the validity of this scale as a screening tool for hikikomori, it may incorrectly identify risk factors or fail to recognize them. The high rate of HLID in this study compared to previous studies also supports this concern.

8. The authors hypothesized that the COVID-19 pandemic would influence the HLID epidemic. Therefore, the authors might want to add the influence of this pandemic on the results of the EFA and CFA. If telework and online learning have become more prevalent and people spend more time at home, would the correlation between the Isolation factor and other factors have been lower than before the pandemic?

9. Lines 534-535

When comparing the characteristics of hikikomori with other studies, the definition of hikikomori should also be compared. For example, in Yong et al. (2020), 47% of the "hikikomori" in this study are full-time workers, although many are certainly married. Since hikikomori are often regarded as those not working (Nonaka et al., 2022), the definition by Yong et al. (2020) differs between these and hikikomori. The high proportion of married people may also just reflect the difference in definitions.

Nonaka, S., Takeda, T., & Sakai, M. (2022). Who are hikikomori? Demographic and clinical features of hikikomori (prolonged social withdrawal): A systematic review. Australian & New Zealand Journal of Psychiatry, 56(12), 1542-1554.

10. There appears to be duplication on lines 185 and 200.

11. There appear to be some inappropriate citations. An overall check is needed. For example, the following is written in line 503: "For example, in a study in China by Yong et al. [88], it was reported that men are more prone to having severe symptoms of HLID compared to females."

However, Yong et al. is a study conducted in Japan. Furthermore, it does not match the list of references.

“89. Yong RK, Fujita K, Chau PY, Sasaki H. Characteristics of and gender difference factors of hikikomori among the working-age population: A cross-sectional population study in rural Japan. Nihon Koshu Eisei Zasshi. 2020;67(4):237-246.”

Reviewer #4: The paper by Al-Sibani et al taps a novel and relevant issue, dealing with the detection of hikikomori(-like idiom of distress, HLID) in general population (of Oman, in the specific case). Authors report on the results of a snow-ball survey, based on the newly translated Arab version of HQ-25, in which 44% of the 1529 subjects expressed HDLI. They conclude on the good validity of the scale and on the possible role of several social and demographic factors may explain the presence of HLID. While the paper is written in a good English and its structure is clear, authors should better explain the recruitment procedure (they report to have recruited 1529 subject, with a high response rate [more than 75%] in the very short period of two weeks). To exclude the risk of sampling bias, Authors should report in more detail within the manuscript the channel they used to publicize the study and all related recruitment procedures.

Reviewer #5: Thank you for the opportunity to review this study that attempts to examine the factor structure of an Arabic language version of the HQ-25 in an internet derived sample from Oman. There is a lot of good material in this paper but unfortunately it has some significant flaws and attempts to achieve more than is justified with the available data. There are some odd analytic choices, most notably the decision to conduct a CFA on exactly the same sample that was used to do the initial EFA. I would encourage the authors to consider re-writing the work as a more focused and psychometrically rigorous cross-cultural scale development study.

The following comments and observations are offered in the hope that they may help improve future versions of the paper.

1. Introduction

The introduction presents a comprehensive and clear account of how factors such as rapid modernisation and economic development combined with society level shocks like the COVID19 pandemic may provoke arise in behavioural phenotypes such as extreme social withdrawal. There is also a good argument made for the choice of the HQ-25 as the measure that is the focus of this study. In many ways, this is the strongest element of the paper.

2. Materials and Methods

Line 206: It seems that the initial sample was obtained via recruitment through social media platforms followed by snowball recruitment. Please provide more information about how the initial point of recruitment worked and any evidence that this ensured an epidemiologically representative sample.

Line 232: The description of the HQ-25 needs careful editing (also noted in the “Typo’s section below). For example, the coefficients for the psychometric properties of the original scale should be presented so that they can be compared to the patterns seen in the current cross-cultural validation of the instrument.

3. Results

Lines 310-312 and Table 1: It is not clear if the information about mental illness reflects a current diagnosis or any lifetime experience of mental illness (that may now be remitted). Please clarify.

Lines 317-366: The authors approach to examining the scale factors was to conduct both an EFA and CFA on the same sample. This violates the standards of good practice for CFA and it is not clear why the authors did not split the sample and perform the EFA first and then attempt CFA on the second half. A clear and rigourous defence of the analytic approach taken is needed before this will be at publishable standard.

Line 386: The column header for Table 4 which states "Yes or No" needs to be edited to make it clear that this refers to falling above or below the cut off for significant Hikikomori features using the HQ-25 cut offs. There should also be a more robust defence of using these cut-offs given that they were derived from scores obtained with non-Omani respondents. The use of the 42/100 cut-off point is mentioned in the discussion but it needs to be defended as part of the methods

4. Discussion

The Discussion is very long and provides a further in-depth narrative that explores why Hikikomori-like Idioms of Distress in Oman may be best seen as a culture reactive phenomenon that is attributable to rapid modernisation, social change, and the impact of the pandemic. Unfortunately, the links back to the data are not as close as they should be and in the analysis of the limitations the authors acknowledge that because they did not measure duration of withdrawal behaviours there is considerable uncertainty about the epidemiological meaning of the presented data. Taking a more focused approach that concentrated on best practice measurement development would have produced a much more defensible paper that could have provided a springboard for next phase studies in Arabic speaking samples (e.g. determining the most sensitive cut-off point on the scale and confirming the factor structure in a second independent sample).

5. Typographical Errors and Areas for Clarification

Lines 195 and 196 – “…this study required at least 1695 to 1958 samples…”. Please replace “samples” with “respondents” (this pattern persists across the entire manuscript – please revise all instances).

Lines 235-236 – Please amend the phrase “HQ-25 is a self-reported instrument of Likert types with a scale ranging from 0…” to reflect that the response options were presented in Likert format.

Line 237 – “The progenitor of this scale ranges from 0-100” – “progenitor” does not make sense in this context – please edit for clarity.

Reviewer #6: The authors present results of a study in Oman that explores the structure of the Hikikomori Questionnaire (HQ-25) translated to Arabic, then confirms the structure and assesses potential risk factors for Hikikomori-like idiom of distress (HLID). Data were acquired through an on-line questionnaire 1 year after the Covid-19 shutdown in Oman. The manuscript will be strengthened if the authors consider the following points.

1. Authors report risk ratios in the abstract, tables and text of the manuscript. However, they have conducted a logistic regression, which provides estimates of odds ratios, not risk ratios. Authors should re-interpret their results in terms of odds rather than probability (or likelihood) and identify the results as odds ratios.

2. It is unclear why authors use a variety of tests/approaches for the univariate analysis when the ultimate goal is to develop a joint logistic regression model - why not just fit simple logistic models with a single predictor and consistently report the odds ratio for all variables? That makes it easier to compare with the results from the multiple logistic regression model.

3. Authors need to be careful in their interpretation of numbers in tables, specifically when noting the subgroup the percentage is referring to. For example, line 310, authors state that of those with HLID, a certain percentage have various mental illness. However, it appears as though the sentence is referring to Table 1 which is for the entire sample. In line 512, authors say of 1035 female participants 71% (n=478) had HLID - 478/1035 is 46.2% (the 71% comes from Table 4, which reports column percentages...so of those with HLID, 71% were female. The same mistake is made in lines 565-566.

4. Table 4: the univariate results do not seem to match the numbers that are presented in the Yes and No columns. For example, for marital status, the result in the table is 6.64, but with the frequencies presented, the OR would be 2.81, while the risk ratio would be 1.73. Some of the reported numbers are actually odds ratios (though they are referred to as risk ratios, which as mentioned in point 1 above, are not the same), while others, I cannot tell how they are calculated, so likely are not either risk ratios or odds ratios.

5. It is not clear why the authors spend an entire paragraph discussing the gender difference, which was only significant in the univariate associations and not in the joint model.

Minor points:

1. line 43: "third is to prevalence" is awkwardly phrased

2. lines 147-148: "Nationwide, ..." is an incomplete sentence

3. line 176: replace "frequency" with "prevalence"

4. lines 201-203 is an incomplete sentence

5. lines 238-239 - authors state that the cut-off of 42 is widely used to determine HLID, but they do not specifically state in the methods that they use this cut-off for the study. I didn't notice specific mention for their study until the discussion.

6. line 271: EPA should be EFA

7. lines 290-291 - the good model fit values for CFI are stated in lines 289-290, so do not need to be restated.

8. line 372: "present of HLID" should be "presence of HLID"

9. lines 400-401: this is an incomplete sentence

10: line 446: "gold-stand" should be "gold-standard"

11. line 618: "less less inflated" should just be "less inflated"

12: line 637: "prevalence rate of HLID" should just be "prevalence of HLID"

7. PLOS authors have the option to publish the peer review history of their article (what does this mean?). If published, this will include your full peer review and any attached files.

Reviewer #1: No

Reviewer #2: No

Reviewer #3: No

Reviewer #4: No

Reviewer #5: No

Reviewer #6: No

---

## [Author Response · Author response to Decision Letter 1]

23 May 2023

Please attached file entitled point counterpoint

---

## [Decision Letter · Decision Letter 2]

7 Jun 2023

PONE-D-22-33971R2Exploring Hikikomori-like idiom of distress a year into the SARS-CoV-2 pandemic in Oman: factorial validity of the 25-item Hikikomori Questionnaire, prevalence and associated factorsPLOS ONE

Dear Dr. Al-Adawi,

Thank you for submitting your manuscript to PLOS ONE. After careful consideration, we feel that it has merit but does not fully meet PLOS ONE’s publication criteria as it currently stands. Therefore, we invite you to submit a revised version of the manuscript that addresses the points raised during the review process.

We look forward to receiving your revised manuscript.

Kind regards,

Boshra Ismael Ahmed Arnout

Academic Editor

PLOS ONE

Journal Requirements:

Reviewers' comments:

Reviewer's Responses to Questions

**Comments to the Author**

1. If the authors have adequately addressed your comments raised in a previous round of review and you feel that this manuscript is now acceptable for publication, you may indicate that here to bypass the “Comments to the Author” section, enter your conflict of interest statement in the “Confidential to Editor” section, and submit your "Accept" recommendation.

Reviewer #2: All comments have been addressed

Reviewer #3: (No Response)

2. Is the manuscript technically sound, and do the data support the conclusions?

Reviewer #2: Yes

Reviewer #3: Yes

3. Has the statistical analysis been performed appropriately and rigorously? 

Reviewer #2: Yes

Reviewer #3: Yes

4. Have the authors made all data underlying the findings in their manuscript fully available?

Reviewer #2: Yes

Reviewer #3: No

5. Is the manuscript presented in an intelligible fashion and written in standard English?

Reviewer #2: Yes

Reviewer #3: Yes

6. Review Comments to the Author

Reviewer #2: 1-the authors should describe the method or technique that was used for setting sampling size in topic 2.4

2-The snowball technique that is not suitable for CFA and inferential statistic that were used in this study.

3-The reseult of this study (in case of CFA) should show or describe the model modification.

verywell revise

Reviewer #3: I believe the authors have adequately addressed many of my previous comments.

However, the following still needed to be adequately addressed.

1. For multicollinearity, a statistic such as VIF should be provided.

2. In the Methods and Discussion of the HQ-25 article by Teo et al. (2018), it is clearly stated that they used the Japanese version (e.g., "Of note, Japanese was used for the administration of the scale." (p. 785)). The authors should carefully check that.

Teo, A. R., Chen, J. I., Kubo, H., Katsuki, R., Sato‐Kasai, M., Shimokawa, N., ... & Kato, T. A. (2018). Development and validation of the 25‐item Hikikomori Questionnaire (HQ‐25). Psychiatry and clinical neurosciences, 72(10), 780-788.

7. PLOS authors have the option to publish the peer review history of their article (what does this mean?). If published, this will include your full peer review and any attached files.

Reviewer #2: No

Reviewer #3: No

---

## [Author Response · Author response to Decision Letter 2]

9 Jun 2023

Please the enclosed entitled point-counterpoint

---

## [Decision Letter · Decision Letter 3]

12 Jul 2023

PONE-D-22-33971R3Exploring Hikikomori-like idiom of distress a year into the SARS-CoV-2 pandemic in Oman: factorial validity of the 25-item Hikikomori Questionnaire, prevalence and associated factorsPLOS ONE

Dear Dr. Al-Adawi,

Thank you for submitting your manuscript to PLOS ONE. After careful consideration, we feel that it has merit but does not fully meet PLOS ONE’s publication criteria as it currently stands. Therefore, we invite you to submit a revised version of the manuscript that addresses the points raised during the review process.

We look forward to receiving your revised manuscript.

Kind regards,

Boshra Ismael Ahmed Arnout

Academic Editor

PLOS ONE

Journal Requirements:

Reviewers' comments:

Reviewer's Responses to Questions

**Comments to the Author**

1. If the authors have adequately addressed your comments raised in a previous round of review and you feel that this manuscript is now acceptable for publication, you may indicate that here to bypass the “Comments to the Author” section, enter your conflict of interest statement in the “Confidential to Editor” section, and submit your "Accept" recommendation.

Reviewer #3: (No Response)

2. Is the manuscript technically sound, and do the data support the conclusions?

Reviewer #3: Partly

3. Has the statistical analysis been performed appropriately and rigorously? 

Reviewer #3: Yes

4. Have the authors made all data underlying the findings in their manuscript fully available?

Reviewer #3: No

5. Is the manuscript presented in an intelligible fashion and written in standard English?

Reviewer #3: Yes

6. Review Comments to the Author

Reviewer #3: The following, which I commented on, still need to be revised in Table 2.

-----

Comment # 5. In Table 2, the data from Teo et al. (2018) are probably the "Japanese" version, not the HQ-25 "English" version.

2. In the Methods and Discussion of the HQ-25 article by Teo et al. (2018), it is clearly stated that they used the Japanese version (e.g., "Of note, Japanese was used for the administration of the scale." (p. 785)). The authors should carefully check that.

------

7. PLOS authors have the option to publish the peer review history of their article (what does this mean?). If published, this will include your full peer review and any attached files.

Reviewer #3: No

---

## [Author Response · Author response to Decision Letter 3]

13 Jul 2023

1. Please include a figure label and title for Figure 1 in your main manuscript 

Authors: Done. Please see lines 361-362 (page 14)

Authors: Thank you. Done as suggested

---

## [Decision Letter · Decision Letter 4]

24 Jul 2023

Exploring Hikikomori-like idiom of distress a year into the SARS-CoV-2 pandemic in Oman: factorial validity of the 25-item Hikikomori Questionnaire, prevalence and associated factors

PONE-D-22-33971R4

Dear Dr. Al-Adawi,

We’re pleased to inform you that your manuscript has been judged scientifically suitable for publication and will be formally accepted for publication once it meets all outstanding technical requirements.

Kind regards,

Boshra Ismael Ahmed Arnout

Academic Editor

PLOS ONE

Additional Editor Comments (optional):

Reviewers' comments:

Reviewer's Responses to Questions

**Comments to the Author**

1. If the authors have adequately addressed your comments raised in a previous round of review and you feel that this manuscript is now acceptable for publication, you may indicate that here to bypass the “Comments to the Author” section, enter your conflict of interest statement in the “Confidential to Editor” section, and submit your "Accept" recommendation.

Reviewer #3: (No Response)

2. Is the manuscript technically sound, and do the data support the conclusions?

Reviewer #3: (No Response)

3. Has the statistical analysis been performed appropriately and rigorously? 

Reviewer #3: (No Response)

4. Have the authors made all data underlying the findings in their manuscript fully available?

Reviewer #3: (No Response)

5. Is the manuscript presented in an intelligible fashion and written in standard English?

Reviewer #3: (No Response)

6. Review Comments to the Author

Reviewer #3: (No Response)

7. PLOS authors have the option to publish the peer review history of their article (what does this mean?). If published, this will include your full peer review and any attached files.

Reviewer #3: No

---

## [Editor Report · Acceptance letter]

28 Jul 2023

PONE-D-22-33971R4 

Exploring Hikikomori-like idiom of distress a year into the SARS-CoV-2 pandemic in Oman: factorial validity of the 25-item Hikikomori Questionnaire, prevalence and associated factors 

Dear Dr. Al-Adawi:

I'm pleased to inform you that your manuscript has been deemed suitable for publication in PLOS ONE. Congratulations! Your manuscript is now with our production department. 

Kind regards, 

on behalf of

Professor Boshra Ismael Ahmed Arnout 

Academic Editor

PLOS ONE